# New International Association for the Study of Lung Cancer (IASLC) Pathology Committee Grading System for the Prognostic Outcome of Advanced Lung Adenocarcinoma

**DOI:** 10.3390/cancers12113426

**Published:** 2020-11-18

**Authors:** Ching-Fu Weng, Chi-Jung Huang, Shih-Hung Huang, Mei-Hsuan Wu, Ailun Heather Tseng, Yung-Chuan Sung, Henry Hsin-Chung Lee, Thai-Yen Ling

**Affiliations:** 1Division of Pulmonary Medicine, Department of Internal Medicine, Hsinchu Cathay General Hospital, Hsinchu 300, Taiwan; cgh18497@cgh.org.tw; 2Department and Graduate Institute of Pharmacology, National Taiwan University, Taipei 100, Taiwan; 3Medical Research Center, Cathay General Hospital, Taipei 106, Taiwan; aaronhuang@cgh.org.tw; 4Department of Biochemistry, National Defense Medical Center, Taipei 114, Taiwan; 5School of Medicine, Fu Jen Catholic University, New Taipei 242, Taiwan; 6Division of Pathology, Cathay General Hospital, Taipei 106, Taiwan; ja68@cgh.org.tw; 7Teaching and Research Center, Hsinchu Cathay General Hospital, Hsinchu 300, Taiwan; markicoo@cgh.org.tw; 8Department of Biomedical Sciences and Engineering, National Central University, Taoyuan 320, Taiwan; c8500@cgh.org.tw; 9Division of Hematology/Oncology, Department of Internal Medicine, Cathay General Hospital, Taipei 106, Taiwan; cgh06496@cgh.org.tw; 10Department of Surgery, Hsinchu Cathay General Hospital, Hsinchu 300, Taiwan; 11Graduate Institute of Translational and Interdisciplinary Medicine, College of Health Sciences and Technology, National Central University, Taoyuan 320, Taiwan

**Keywords:** late stage, lung adenocarcinoma, histology, subtype, EGFR mutation, smoker, sex

## Abstract

**Simple Summary:**

This study investigated the association between survival outcome and the new grading system among advanced stage lung adenocarcinoma (LADC) (stages IIIA, IIIB and IV) patients who were diagnosed as LADC with a pathologic report according to a new grading system by the International Association for the Study of Lung Cancer (IASLC) pathology committee. The results indicate that the poorly differentiated group had a poorer prognosis in PFS, as did patients with wild-type EGFR who were treated with chemotherapy. No survival difference could be found among EGFR mutation patients. Older age and a lower body mass index also led to worse survival. Patients with poorly differentiated adenocarcinoma likewise had worse survival, especially compared to those with moderately differentiated adenocarcinoma. Our findings highlight that the therapeutic regimen should be adjusted for wild-type EGFR patients with poorly differentiated adenocarcinoma treated with chemotherapy to provide better outcomes. No survival difference could be seen among EGFR mutation patients.

**Abstract:**

The impact of the new International Association for the Study of Lung Cancer pathology committee grading system for advanced lung adenocarcinoma (LADC) on survival is unclear, especially in Asian populations. In this study, we reviewed the prognostic outcomes of patients with late-stage disease according to the new grading system. We reviewed 136 LADC cases who underwent a small biopsy from 2007 to 2018. Tumors were classified according to the new grading system for LADC. Baseline characteristics (age, sex, smoking status, body mass index, and driver gene mutations) were analyzed. Kaplan–Meier and Cox regression analyses were used to determine correlations with the new grading system and prognosis. Patients with poorly differentiated adenocarcinoma were significantly correlated with a poor progression-free survival (PFS) (*p* = 0.013) but not overall survival (OS) (*p* = 0.154). Subgroup analysis showed that wild-type EGFR patients with poorly differentiated adenocarcinoma treated with chemotherapy had significantly worse PFS (*p* = 0.011). There was no significant difference in survival among the patients with epidermal growth factor receptor mutations who were treated with tyrosine kinase inhibitors. Patients aged >70 years and those with a BMI ≤ 25 kg/m^2^ and wild-type patients had significantly worse OS in both univariate (HR = 1.822, *p* = 0.006; HR = 2.250, *p* = 0.004; HR = 1.537, *p* = 0.046, respectively) and multivariate analyses (HR = 1.984, *p* = 0.002; HR = 2.383, *p* = 0.002; HR = 1.632, *p* = 0.028, respectively). Despite therapy, patients with poorly differentiated tumors still fared worse than those with better differentiated tumors. No differences were found among the EGFR mutations treated with TKI. Our findings highlight that the therapeutic regimen should be adjusted for EGFR Wild-type patients with poorly differentiated adenocarcinoma treated with chemotherapy to provide better outcomes.

## 1. Introduction

Lung cancer is a leading cause of life-threatening malignancy worldwide. According to the revised classification criteria released by the International Association for the Study of Lung Cancer (IASLC), the American Thoracic Society (ATS), and the European Respiratory Society (ERS) in 2011, lung adenocarcinoma (LADC) can be categorized into five subtypes (lepidic, acinar, solid, papillary, and micropapillary) [1]. Most previous cohort studies were based on the IASLC/ATS/ERS classification, which is used to predict outcomes in the early stages of the disease [2,3,4,5,6,7]. However, Moreira et al. proposed a new grading system for invasive pulmonary adenocarcinoma taking into account the heterogeneity of pulmonary adenocarcinomas that involves useful prognostic groupings based on the predominant and high-grade histologic subtypes [8].

To date, several studies have investigated the impact of the IASLC/ATS/ERS classification system introduced in 2011 to predict the prognosis in patients with advanced LADC, although tumor heterogeneity in late-stage LADC is widely known to be complex [9,10,11,12]. Campos-Parra et al. reported that high-grade LADC (micropapillary, papillary, and solid-predominant) was associated with better outcomes compared to intermediate-grade LADC (lepidic- and acinar-predominant), especially when treated with standard platinum-based doublet chemotherapy [12]. Da Cruz et al. concluded that predominant subtyping is reliable in the prediction stage for IV LADC, especially in solid subtype [10], whereas Clay et al. indicated that the major solid histologic subtype of metastatic LADC is associated with inferior survival outcomes under systemic treatment [9]. Small biopsies remain the primary method for diagnosis, classification, and molecular analysis, accounting for 70% in advanced disease [13]. As the histologic subtype is an important predictive factor, core biopsy results are reliable representations of the original tumor entity and valuable for predicting prognosis [14,15,16,17,18]. In addition, it is vital to assess the relevance of driver gene mutations, such as the epidermal growth factor receptor (EGFR), anaplastic lymphoma kinase (ALK), Kirsten rat sarcoma (KRAS), and proto-oncogene B-Raf (BRAF), in different subtypes when predicting a patient’s prognosis [19,20,21,22].

To evaluate the prognostic relevance of the new grading system [8], we aimed to apply this system to patients with advanced LADC harboring (or not) driver gene mutations and analyze the potential prognostic differences. In this study, we examined the clinical relevance for Asian patients with late-stage disease using small biopsy samples.

## 2. Results

### 2.1. Clinicopathological Factors

As classified by the IASLC pathology committee’s new grading system, 136 LADC cases out of a total of 1317 met the inclusion criteria and were analyzed (Figure 1). Of the 136 patients, 7 (5.1%) had well differentiated adenocarcinoma, 74 (54.4%) had moderately differentiated adenocarcinoma, and 55 (40.4%) had poorly differentiated adenocarcinoma (Figure 1 and Figure 2 and Table 1). There were 71 (52.2%) females and 65 (47.8%) males in the sample, ranging in age from 31 to 91 years (mean = 65.3 years), and 87 (63.5%) patients were aged <70 years. In total, 92 patients (67.6%) were never smokers, and 93 patients (72.1%) had a BMI ≤ 25 kg/m^2^. Fifty-eight (42.6%) patients had wild-type EGFR, and 78 (57.4%) had driver gene mutations. Seventy-six (69.7%) patients received a first-line treatment with TKIs, and 33 (30.3%) received platinum-based chemotherapy. Sex, BMI, smoking status, stage, and treatment were all independent of the grade (*p* > 0.05, chi-square test) (Table 2).

### 2.2. EGFR Wild-Type and EGFR Mutations and First-Line Treatment

EGFR mutations were also independent of grade (*p* > 0.05, chi-square test). There were 58 wild-type EGFR patients (42.6%), and 78 (57.4%) had EGFR mutations. Among those with EGFR mutations, there were 7 exon 18 mutations (5.1%), 33 exon 19 deletions (24.3%), 2 exon 20 insertions (1.5%), and 34 exon 21 point mutations (25.0%). Only two patients had EML4-ALK fusions (1.5%). In the poorly differentiated group, 29 (21.3%) patients had EGFR mutations, compared to 45 (33.1%) in the moderately differentiated group and 4 (2.9%) in the well differentiated group. Most of the mutations were exon 19 deletions and exon 21 point mutations. The frequency of gene mutations was slightly higher in the moderately differentiated group, which was also detected in those receiving first-line TKIs (44, 40.4%) (Table 2).

### 2.3. Survival Analysis

The survival outcomes of the poorly differentiated, moderately differentiated, and well differentiated groups diagnosed between January 2007 and December 2018 were analyzed. Each group had a statistically significant difference in PFS (*p* = 0.013) but not in OS (*p* = 0.154) (Figure 3A,B). Those without systemic treatment (*n* = 27, Table 2) had a predominantly inferior outcome in both PFS (*p* < 0.001) and OS (*p* < 0.001) (Appendix A). No difference in PFS or OS was seen between each group (Appendix A). When the survival analysis examined only wild-type EGFR patients, the poorly differentiated group had the worst PFS (*p* = 0.011) when treated with chemotherapy (Figure 4A,B). This was not observed in those with EGFR mutations when treated with TKIs (Appendix A).

We found no differences in the prognostic outcomes with regards to age, sex, or smoking status. Poorer overall survival was detected in the patients with a BMI ≤ 25 kg/m^2^ and age >70 years in late-stage disease (Appendix A). In the univariate analysis, those aged >70 years, those with a BMI ≤ 25, and wild-type EGFR patients were significantly associated with a worse OS (HR = 1.822, *p* = 0.006; HR = 2.250, *p* = 0.004; HR = 1.537, *p* = 0.046, Table 3). Similar results were also detected in the multivariate analysis (HR = 1.984, *p* = 0.002; HR = 2.383, *p* = 0.002; HR = 1.632, *p* = 0.028, Table 3). In particular, the poorly differentiated group had a worse PFS compared to the moderately differentiated group (Table 4). The Kaplan–Meier analysis provided the same results for older age, lower BMI, and wild-type EGFR patients (Table 5).

In brief, poorly differentiated patients had the worst PFS in this study. In the subgroup analysis, the poorly differentiated group with wild-type EGFR also had a worse PFS when treated with chemotherapy.

## 3. Discussion

The classification of patients with LADC has been addressed in previous studies, and valuable prognostic features have been identified according to the revised IASLC/ATS/ERS system released in 2011. The importance in early stage is well-documented [1,2,4,6,23,24,25,26]. The predominant type of LADC can be used to predict survival after adjuvant chemoradiotherapy, which represents OS in low-risk groups with a reported OS of 78.5 months for lepidic-predominant LADC, 67.3 months for intermediate-grade (acinar-predominant) LADC, and 57.2 months for high-grade (papillary, micropapillary, and solid-predominant) LADC. In this study, the patients with papillary-predominant LADC had an equivalent survival rate to the patients with micropapillary- and solid-predominant LADC [27]. Similar results have also been reported in other cohorts. It is generally accepted that patients with solid and/or micropapillary patterns have a worse prognosis whether or not the subtype is predominant [28,29,30].

The previous limited resected sample of LADC was able to predict poor prognostic outcomes in early-stage disease. The new grade system proposed by Moreira et al. provided a more complete scale for redefining the predominant subtyping into different categories for predicting the prognosis [8]. Although the clinical relevance of major histologic patterns in late-stage disease has been addressed, the specimens in these studies were mainly acquired by surgical resection or open biopsy [9,10]. Whether a small biopsy is representative of the actual predominant subtypes needs verification using a larger cohort [15]. Since most patients were diagnosed using small biopsy specimens in our study, a validation and correlation of grading between biopsies and resection is warranted. The new grading system in our research is applicable for two reasons. First, small biopsies and cytology specimens from primary or metastatic sites were verified by Sørensen et al. according to the 1981 WHO classification [31]. Second, despite the heterogenous nature of advanced LADC [13,25,32] (also detected in small biopsy specimens [14,15,16,17,18], a cutoff of 20% solid or micropapillary patterns was proposed to determine metastatic potential [3,8]. Clay et al. also noted that a solid pattern is the most frequent pattern in metastatic LADC, and 48% patients in this study were diagnosed by either open biopsy or core biopsy [9]. Therefore, we surmise that biopsy in late-stage LADC and a critical cutoff of 20% solid or micropapillary patterns could provide a strong indicator of prognosis.

The previous limited resected sample sizes of LADC were able to predict poor prognostic outcomes in early-stage disease, but the role of each subtype in predicting the prognosis during the late stage remains unclear [2,9,14,15,29]. In the current study, we showed that the use of the new grading system in patients with advanced stage disease is reproducible and applicable in clinical practice. More than 70% of our patients were diagnosed with advanced-stage lung cancer, which is usually proven via small biopsies or cytology in late-stage disease. The finding that most of the tumors were of a predominant pattern represents the heterogenous nature of tumors in advanced LADC [13,19,25,32]. On the other hand, Moreira et al. and Sica et al. proposed that a cutoff of 20% solid or micropapillary patterns can indicate metastatic potential [3,8] in early-stage disease. Thus, poorly differentiated adenocarcinoma with more than a 20% solid or micropapillary dominant patterns could provide a strong indicator of poor prognosis for patients compared with other groups (moderately differentiated and well differentiated).

Another important issue is that driver gene mutations are highly expressed in certain subtypes [33,34,35]. In patients with advanced-stage LADC harboring EGFR mutations, TKIs are efficient and widely used as the first-line therapeutic choice. Nevertheless, the duration of PFS and OS ranges widely according to the individuals. Cancer stem cell formation with tumor heterogeneity, DNA and epigenetic changes, transcriptome or signal pathway alterations, gene copy number variations, and chromosomal instability may all contribute to drug resistance and eventually cause treatment failure [20].

In this study, the poorly differentiated group had the worst PFS, and this result was also seen in the subgroup analysis of those with the wild-type EGFR receiving chemotherapy. There were no significant differences in sex and smoking status. Those aged >70 years and those with a BMI ≤ 25 kg/m^2^ had a worse OS. This highlights that those with poorly differentiated adenocarcinoma might fare worse than the other groups with systemic chemotherapy, which is similar to previous research showing that major solid patterns indicate inferior OS after systemic treatment [9]. Nevertheless, this result seems to conflict with other studies’ conclusions [36,37,38,39]. It is possible that different regimens may influence survival, especially as the method of chemotherapy can vary over time [29]. Therefore, we still provided possible predictors of preterm treatment failure in patients receiving first-line chemotherapy based on the histologic predominant subtype. Earlier treatment strategies should be tailored to improve the prognosis of patients with certain characteristics. For example, chemotherapy plus anti-angiogenic drugs or immunotherapy might be choices for active disease control and improving drug efficacy. Further studies are also needed to examine the efficacy of combination therapy. No significant differences were found among patients with EGFR mutations who received first-line EGFR TKIs in this study, which highlights the important role of molecular staging and cancer genomics in future precision medicine [20,40]. To date, relatively little progress has been made in identifying patients who are at risk of relapse after surgical resection or the metastatic process based on the characteristics of advanced-stage lung cancer. Understanding the surveillance of immunotherapy and the microenvironment targeting neoantigens may provide new insight into relevant treatment strategies. Our results indicate that pathologic subtyping in patients with LADC can provide guidance in wild-type EGFR patients, whereas a genomic survey is crucial for targeted therapy.

There are several limitations to the current study. First, the relatively small sample size may have affected the interpretation of the results of the survey. This could also have generated selection bias when performing the statistical analysis, so the results must be validated in a larger cohort in the future. Second, the tumor samples were acquired from small biopsies or surgical resection, and this may have affected the accuracy of the pathological interpretation; however, this remains the standard method for diagnosis in late-stage disease. The tumor heterogeneity in late-stage disease highlights the complexity of the tumor, and different approaches for sample acquisition could eliminate sampling errors. Third, around 30.3% of the wild-type EGFR patients received first-line chemotherapy. This relatively low treatment rate could also have led to a potential bias during analysis, even though our results were statistically significant.

## 4. Materials and Methods

### 4.1. Participants

From January 2007 to December 2018, patients with late-stage (stage IIIA, IIIB and IV) LADC who received treatment at Cathay General Hospital (CGH), Taipei, Taiwan were identified. The inclusion criteria for the current study were as follows: (i) patients who received a lung tumor biopsy at CGH (FFPE: formalin-fixed paraffin-embedded tissues obtained through computed tomography (CT)-guided needle biopsy, echo-guided core needle biopsy, or transbronchial biopsy); and (ii) pathological confirmation of LADC using the new grading system [8]. The exclusion criteria were patients: (i) who were lost to follow-up; (ii) who received incomplete treatment; (iii) who received a formal pathology report from another hospital; (iv) who were diagnosed pathologically through a pleural effusion cell block; (v) for whom no definite pathological subtype was identified; (vi) for whom the carcinoma was of uncertain origin, adenosquamous, or neuroendocrine; and (vii) with unknown gene mutation status. This study was approved by Cathay General Hospital (CGH), Taipei, Taiwan (NO.: CGH-P108001). The informed consent form was waived in this study.

All biopsy specimen interpretations under the new grading system based on the IASLC pathology committee classified the adenocarcinomas as being well differentiated, moderately differentiated, or poorly differentiated. The grading scheme in this model comprised lepidic predominant patterns with a 20% or lower cutoff of high-grade patterns categorized as the well differentiated group, while acinar or papillary predominant patterns with or less than 20% of high-grade patterns comprised the moderately differentiated group. Finally, any tumor with 20% or more high-grade patterns was defined as poorly differentiated [8].

We also evaluated other parameters including sex, age at diagnosis, smoking history, body mass index (BMI), gene mutation status, and staging according to the eighth edition of the lung cancer staging system [41]. The first-line treatment regimen used, progression-free survival (PFS), and overall survival (OS) were recorded based on the follow-up medical records. The current study was approved by the Institutional Review Board of Cathay General Hospital (Approval No. CGH-P108001).

### 4.2. Mutational Analysis

In total, 137 FFPE lung malignant tissue samples were collected by CT-guided or echo-guided biopsy from the Department of Pathology, CGH. Mutations of *EGFR*, *KRAS,* and *BRAF* were determined following standard protocols (EGFR Mutation Test V2, KRAS Mutation Test V2, and BRAF/ NRAS Mutation Test (LSR); Roche, Mannheim, Germany). ALK fusions (D5F3; Roche, Mannheim, Germany) were confirmed by immunohistochemistry.

### 4.3. Statistical Analysis

The statistical analysis was performed using SPSS version 20.0 (IBM Corp, Armonk, NY, USA). Correlations between two categorical variables were examined using Pearson’s chi-squared test. PFS and OS were calculated at 12 and 36 months after the initial diagnosis, respectively. The Kaplan–Meier method was used to analyze the distributions of PFS and OS, and log-rank tests were performed to compare the differences between two categories. Univariate analysis and multivariate survival analysis were conducted using Cox proportional hazard regression to obtain the hazard ratios (HRs) with 95% confidence intervals (CIs) and identify independent prognostic factors for PFS and OS. Statistical significance was set at α = 0.05.

## 5. Conclusions

In conclusion, this is the first study to apply the new grading system of the IASLC for predicting the survival prognosis of Asian patients with advanced LADC. Negative impacts were shown on PFS and OS in certain patient subgroups, including poorly differentiated patients, those with a lower BMI, and the elderly. Despite therapy, patients with poorly differentiated tumors still fared worse than those with better differentiated tumors. Conventional first-line chemotherapy treatment with combination therapy using anti-angiogenic agents or immunotherapy might be helpful for improving treatment outcomes. Further large-scale studies are needed to establish whether this treatment strategy is valid.

## Figures and Tables

**Figure 1 cancers-12-03426-f001:**
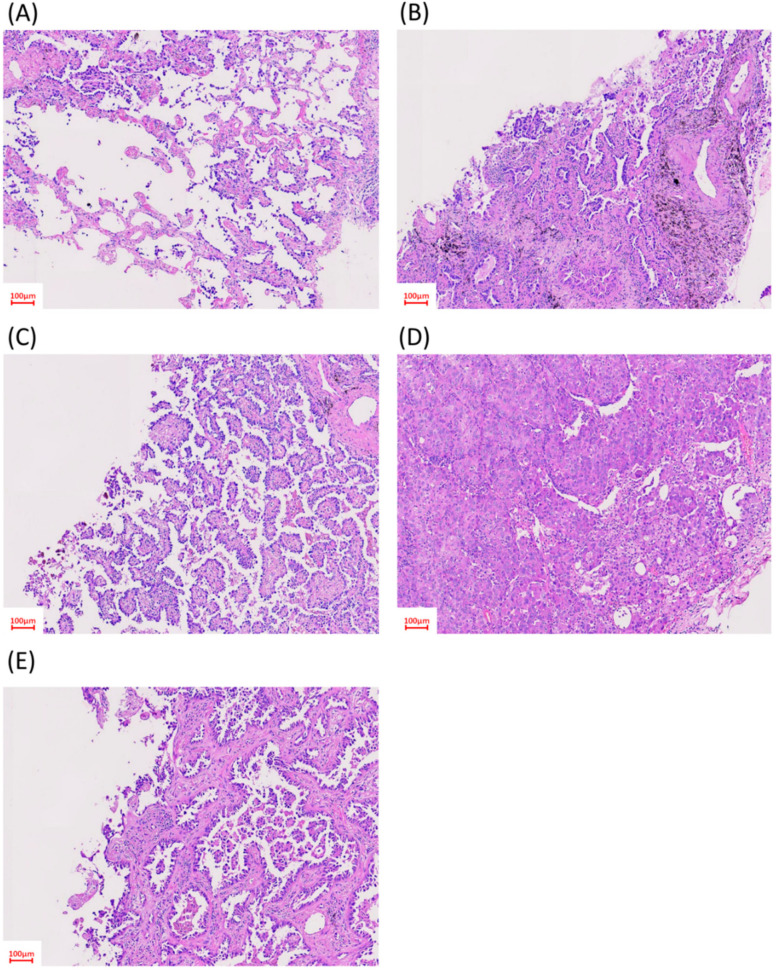
Histological patterns of adenocarcinoma. Representative hematoxylin and eosin-stained biopsy specimens: (**A**) lepidic-predominant pattern; (**B**) acinar-predominant pattern; (**C**) papillary-predominant pattern; (**D**) solid-predominant pattern; and (**E**) micropapillary-predominant pattern.

**Figure 2 cancers-12-03426-f002:**
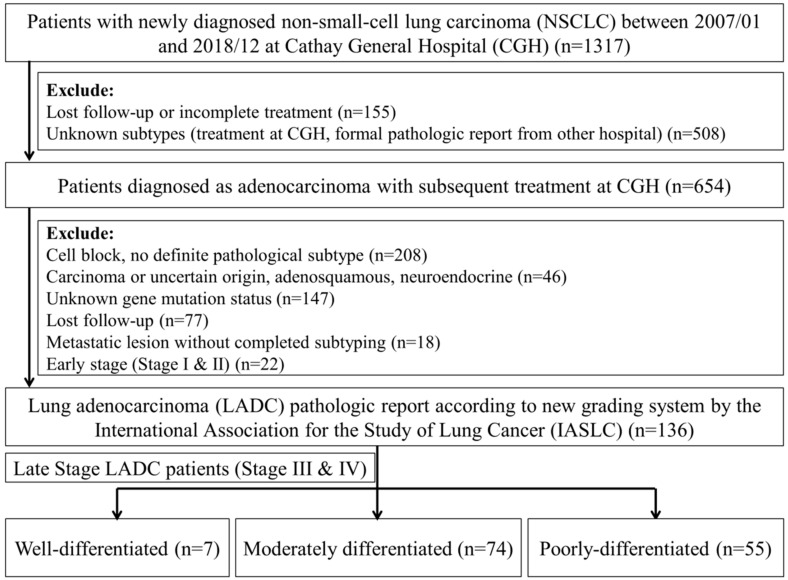
Selection criteria for the subjects.

**Figure 3 cancers-12-03426-f003:**
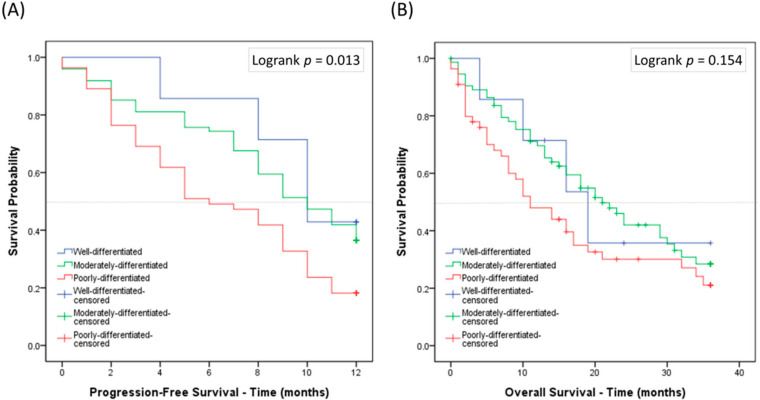
The Kaplan–Meier survival curves for each group of the new grading system: (**A**) progression-free survival; and (**B**) overall survival.

**Figure 4 cancers-12-03426-f004:**
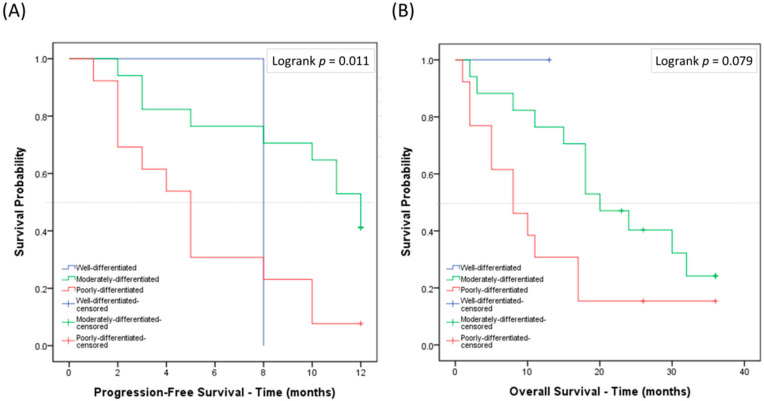
Kaplan–Meier survival curves for each group with wild-type EGFR and those treated with chemotherapy under new grading system: (**A**) progression-free survival; and (**B**) overall survival. Patient numbers in the respective groups are listed in Table 1.

**Table 1 cancers-12-03426-t001:** Adenocarcinoma subtyping according to the New IASLC grading system.

Grade	Differentiation	N (%)
Grade 1	Well differentiated	7 (5.1%)
Grade 2	Moderately differentiated	74 (54.4%)
Grade 3	Poorly differentiated	55 (40.5%)
Total number of included patients: 136		

**Table 2 cancers-12-03426-t002:** Relationships among differentiation and clinicopathological variables in advanced lung adenocarcinoma patients.

Variables	N (%)	Well Differentiated	Moderately Differentiated	Poorly Differentiated	*p* Value
		7 (5.1%)	74 (54.4%)	55 (40.4%)	
Sex					
Male	65 (47.8%)	3 (2.2%)	31 (22.8%)	31 (22.8%)	0.257
Female	71 (52.2%)	4 (2.9%)	43 (31.6%)	24 (17.6%)	
Age groups (years)					
≤70	87 (64.0%)	7 (5.1%)	50 (36.8%)	30 (22.1%)	0.039
>70	49 (36.0%)	0 (0.0%)	24 (17.6%)	25 (18.4%)	
BMI (kg/m^2^)					
≤25	93 (72.1%)	5 (3.9%)	50 (38.8%)	38 (29.5%)	0.979
>25	36 (27.9%)	2 (1.6%)	20 (15.5%)	14 (10.9%)	
Missing	7	0	4	3	
Smoking Status					
Non-smoker	92 (67.6%)	6 (4.4%)	49 (36.0%)	37 (27.2%)	0.572
Ever smoker	44 (32.4%)	1 (0.7%)	25 (18.4%)	18 (13.2%)	
Stage					
IIIA & IIIB	17 (12.5%)	1 (0.7%)	9 (6.6%)	7 (5.1%)	0.985
IV	119 (87.5%)	6 (4.4%)	65 (47.8%)	48 (35.3%)	
Driver gene mutation					
EGFR Wild-type	58 (42.6%)	3 (2.2%)	29 (21.3%)	26 (19.1%)	0.656
EGFR Mutation	78 (57.4%)	4 (2.9%)	45 (33.1%)	29 (21.3%)	
Exon 18 mutation	7 (5.1%)	1 (0.7%)	5 (3.7%)	1 (0.7%)	
Exon 19 deletion	33 (24.3%)	2 (1.5%)	16 (11.8%)	15 (11.0%)	
Exon 20 insertion mutation	2 (1.5%)	0 (0.0%)	2 (1.5%)	0 (0.0%)	
Exon 21 point mutation	34 (25.0%)	1 (0.7%)	21 (15.4%)	12 (8.8%)	
EML4-ALK	2 (1.5%)	0 (0.0%)	1 (0.7%)	1 (0.7%)	
Treatments					
TKIs	76 (69.7%)	5 (4.6%)	44 (40.4%)	27 (24.8%)	0.650
Chemotherapy	33 (30.3%)	1 (0.9%)	18 (16.5%)	14 (12.78%)	
Not systemic treatment	27	1	12	14	

Tyrosine kinase inhibitor (TKI), gefitinib (Iressa), erlotinib (Tarceva), afatinib (Giotrif), and crizotinib; chemotherapy, pemetrexed (Alimta), gemcitabine (Gemzar), and vinorelbine (Navelbine); *p*-values from Pearson’s chi-squared test. BMI: body mass index; EGFR: epidermal growth factor receptor; EML4: echinoderm microtubule-associated protein-like 4; ALK: anaplastic lymphoma kinase;

**Table 3 cancers-12-03426-t003:** Univariate and multivariate analyses of the overall survival in advanced lung adenocarcinoma patients.

Variables	OS					
	Univariate			Multivariate		
	HR	95% CI	*p* Value	HR	95% CI	*p* Value
Groups						
Well differentiated	1					
Moderately differentiated	1.030	0.370–2.866	0.955			
Poorly differentiated	1.548	0.552–4.340	0.406			
Age groups (years)						
≤70	1			1		
>70	1.822	1.183–2.807	0.006 **	1.984	1.273–3.093	0.002 **
Sex						
Female	1					
Male	1.039	0.841–1.285	0.721			
Smoking Status						
Non-smoker	1					
Ever-Smoker	0.956	0.607–1.506	0.846			
BMI group (kg/m^2^)						
>25	1			1		
≤25	2.250	1.297–3.906	0.004 **	2.383	1.372–4.139	0.002 **
EGFR						
Mutation	1			1		
Wild-type	1.537	1.008–2.344	0.046 *	1.632	1.055–2.523	0.028 *
Treatments						
TKI	1					
Chemotherapy	1.538	0.939–2.520	0.087			

OS, overall survival. Note: Missing values are excluded. TKI: gefitinib (Iressa), erlotinib (Tarceva), afatinib (Giotrif), and crizotinib. Chemotherapy: pemetrexed (Alimta), gemcitabine (Gemzar), and vinorelbine (Navelbine). HR: hazard ratio, ** *p* < 0.01, * *p* < 0.05.

**Table 4 cancers-12-03426-t004:** Factors associated with progression-free survival (PFS) under chemotherapy.

Variables (N)	PFS Months	Univariate		
		HR	95% CI	*p* Value
Well differentiated (3)	10 (6.8–13.2)	1		
Moderately differentiated (29)	11 (8.4–13.6)	0.952	0.219–4.142	0.948
Well differentiated (3)	10 (6.8–13.2)	1		
Poorly differentiated (26)	4 (2.5–5.5)	2.897	0.676–12.415	0.152
Moderately differentiated (29)	11 (8.4–13.6)	1		
Poorly differentiated (26)	4 (2.5–5.5)	2.800	1.495–5.352	0.002 **

Data are presented as the median (95% CI). HR, hazard ratio. ** *p* < 0.01.

**Table 5 cancers-12-03426-t005:** Progression-free survival and overall survival according to baseline characteristics.

Variables	Number	Median PFS (Months)	*p* Value ^a^	Median OS (Months)	*p* Value ^a^
Groups					
Well differentiated	7	10 (7.434–12.566)	0.013 **	19 (9.224–28.776)	0.154
Moderately differentiated	74	10 (8.057–11.943)		21 (15.527–26.473)	
Poorly differentiated	55	6 (3.358–8.642)		11 (6.106–15.894)	
Age groups (years)					
≤70	87	10 (8.945–11.055)	0.118	21 (15.912–26.088)	0.005 **
>70	49	7 (4.441–9.559)		11 (6.003–15.997)	
Sex					
Male	65	8 (5.637–10.363)	0.144	13 (8.755–17.245)	0.716
Female	71	9 (7.731–10.269)		19 (14.818–23.182)	
BMI group (kg/m^2^)					
≤25	93	9 (7.903–10.097)	0.222	16 (12.009–19.991)	0.003 **
>25	36	10 (7.652–12.348)		34	
Smoking Status					
Non-smoker	92	9 (7.958–10.042)	0.384	19 (13.620–24.380)	0.843
Ever-Smoker	44	8 (6.927–9.073)		17 (12.180–21.820)	
EGFR					
Wild-type	58	6 (3.512–8.488)	0.131	13 (8.285–17.715)	0.041 *
Mutation	78	9 (7.848–10.152)		21 (10.759–31.241)	
EGFR-m ^b^					
Sex					
Male	34	9 (6.718–11.282)	0.723	29 (9.052–48.948)	0.696
Female	44	9 (7.701–10.299)		21 (17.448–24.552)	
EGFR-m & Non-smoker					
Sex					
Male	20	9 (6.820–11.180)	0.481	14 (9.866–18.134)	0.913
Female	41	10 (8.754–11.246)		21 (17.429–24.571)	
Treatments					
TKI ^c^	76	10 (8.938–11.062)	0.140	22 (16.760–27.240)	0.080
Chemotherapy ^d^	33	8 (3.981–12.019)		17 (10.728–23.272)	

^a^ Log rank test; ^b^ EGFR-m, EGFR mutation; ^c^ TKI, gefitinib (Iressa), erlotinib (Tarceva), afatinib (Giotrif), and crizotinib; ^d^ chemotherapy, pemetrexed (Alimta), gemcitabine (Gemzar), and vinorelbine (Navelbine); PFS, progression-free survival; OS, overall survival. Note: Missing values are excluded. ** *p* < 0.01, * *p* < 0.05.

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
