# Peer review of "New International Association for the Study of Lung Cancer (IASLC) Pathology Committee Grading System for the Prognostic Outcome of Advanced Lung Adenocarcinoma"

_cancers, 2020, doi:10.3390/cancers12113426_

Round 1
Reviewer 1 Report
The authors evaluate the new proposed grading system for adenocarcinoma in advanced stage lung cancer patients. The concluded that the grading system is applicable to advanced stages but does not seem to interfere with outcome post chemotherapy.
1- the grading system was developed in surgical resection specimens. it has not been validated in small biopsies. it would be very important if the authors show first a validation or correlation of grading between biopsies and resection, before the evaluation of advances stages. Although, the authors cite a reference where patterns evaluation have show concordance between resection and biopsy, there is no data on grading.
2- the number of cases evaluated is really very small for any conclusion of benefit or influence of grading in response to therapy. Their conclusion that poorly-differentiated tumors treated with chemotherapy have worse prognosis is not valid. they do not have a control group to evaluate therapy. all they can conclude is that despite therapy, patients with poorly differentiated tumor still fair worse that those with better differentiated tumors.
3-Table 4, how the authors explain that there in difference in PFS in poorly differentiated tumor compared to well-differentiated tumors?, but yet there is a difference when compared with moderately differentiated?
4- how they explain the lack of correlation between grade and outcome in patients with EGFR mutation?
5-The number in the table do not seem to match. It was stated that there were only 33 patients without GFR mutation, but the table seems to suggest a different number.
6- Discussion: line 228 " poorly-differentiation could potentially hamper..." This is not supported by their data.
Author Response
Dear Reviewer:
Thank you for the comments regarding our manuscript. Below, you will find a point-by-point response to these comments. We believe this round of critical review now meets the level of excellence needed for publication in Cancers.
Sincerely,
Thai-Yen Ling

Reviewer 2 Report
on this study the authors reviews the prognostic outcomes of patients with late-stage disease according to the new grading system. Tumors were classified according to the new grading system for lung adenocarcinoma (LADC). Patients with poorly differentiated adenocarcinoma were significantly correlated with poor progression-free survival (PFS) (p=0.013), but not overall survival (OS)(p=0.154). Wild-type patients with poorly differentiated adenocarcinoma treated with chemotherapy had significantly worse PFS (p=0.011). Patients with poorly differentiated adenocarcinoma had worse survival, especially when compared to those with moderately differentiated adenocarcinoma. The findings highlight that the therapeutic regimen should be adjusted in wild-type patients with poorly differentiated adenocarcinoma treated with chemotherapy for better outcomes.
line 123, replaced GFR by EGFR
Author Response

(The authors gave the same response as above.)

Reviewer 3 Report
The authors apply the newly proposed IASLC grading system to a cohort of advanced stage lung cancer patients from Taiwan and determine that poor differentiation is associated with worse progression free survival in “wild type” patients. Analysis of grade in an advanced stage cohort like this is interesting but tricky given the confounding by variable therapeutic intervention and use of targeted therapy. The authors address these confounders by applying multivariate analyses. The survival analyses suggest that grade is associated with PFS but not OS among the overall cohort and in the chemotherapy treated cohort. Grade had no impact on outcomes among EGFR TKI treated patients.
In the simple summary and abstract, the authors should better define “wild type patients”. EGFR wild type? Or pan “targetable” biomarker wild type?
Methods: what proportion of specimen is resection vs biopsy? Given that the IASLC grading criteria were developed and validated on resection specimens, the use of biopsies introduces another variable that needs to be accounted for.
Results:
Table 1: rather than mixed subtypes (which sounds like a typical adenocarcinoma pattern), perhaps the authors mean mixed types (presumably these are combined adenocarcinoma and small cell carcinoma or other?)
Figure 1- it is not clear that panel C shows papillary growth. This looks lepidic and maybe acinar with some exuberant epithelial proliferation but no papillary cores. Panel E is not convincingly micropapillary-predominant given what appears to be frequent papillary cores.
Figures 3 and 4 legends should be more detailed- one can only discern the difference by reading the text.
Figure 5 is of limited relevance to the focus of the study and would be better placed in the supplement. the EGFR TKI curves may be negative but are more pertinent to the conclusions.
Discussion:
The first paragraph states that papillary predominant tumors have survival rates on par with micropapillary and solid tumors, however this data is not provided in the results text. Please also see comment above regarding figure 1 and obtain additional expert opinions about the classification of papillary and micropapillary-predominant tumors.
Author Response

(The authors gave the same response as above.)

Round 2
Reviewer 3 Report
The authors have adequately addressed my concerns.
Author Response
Dear Reviewer,
Thank you so much for valuable comments. I appreciated and we had also sent our manuscript for English editing by MDPI editing system. We believe the revised version meets the excellence to publish in Cancers. Wish you have a good day.
Sincerely,
Ching-Fu Weng
